# Papillary Thyroid Cancer Prognosis: An Evolving Field

**DOI:** 10.3390/cancers13215567

**Published:** 2021-11-07

**Authors:** Salvatore Ulisse, Enke Baldini, Augusto Lauro, Daniele Pironi, Domenico Tripodi, Eleonora Lori, Iulia Catalina Ferent, Maria Ida Amabile, Antonio Catania, Filippo Maria Di Matteo, Flavio Forte, Alberto Santoro, Piergaspare Palumbo, Vito D’Andrea, Salvatore Sorrenti

**Affiliations:** 1Department of Surgical Sciences, “Sapienza” University of Rome, 00161 Rome, Italy; enke.baldini@uniroma1.it (E.B.); augusto.lauro@uniroma1.it (A.L.); daniele.pironi@uniroma1.it (D.P.); domenico.tripodi@uniroma1.it (D.T.); eleonora.lori@uniroma1.it (E.L.); Iulia.ferent@uniroma1.it (I.C.F.); mariaida.amabile@uniroma1.it (M.I.A.); antonio.catania@uniroma1.it (A.C.); filippomaria.dimatteo@uniroma1.it (F.M.D.M.); alberto.santoro@uniroma1.it (A.S.); piergaspare.palumbo@uniroma1.it (P.P.); vito.dandrea@uniroma1.it (V.D.); salvatore.sorrenti@uniroma1.it (S.S.); 2Urology Department, M.G. Vannini Hospital, 00177 Rome, Italy; flavioforte@hotmail.com

**Keywords:** thyroid cancers, molecular pathogenesis, prognosis, therapy, TNM, tumor molecular profiling, BRAF, TERT promoter, plasminogen activating system, miRNA, estrogen receptor

## Abstract

**Simple Summary:**

Over the last couple of decades, the prognostic stratification systems of differentiated thyroid cancer (DTC) patients have been revised several times in an attempt to achieve a tailored clinical management reflecting the single patients’ needs. Such revisions are likely to continue in the near future, since the prognostic value of a number of promising clinicopathological features and new molecular biomarkers are being evaluated. Here, we will review the current staging systems of thyroid cancer patients and discuss the most relevant clinicopathological parameters and new molecular markers that are potentially capable of refining the prognosis.

**Abstract:**

Over the last few years, a great advance has been made in the comprehension of the molecular pathogenesis underlying thyroid cancer progression, particularly for the papillary thyroid cancer (PTC), which represents the most common thyroid malignancy. Putative cancer driver mutations have been identified in more than 98% of PTC, and a new PTC classification into molecular subtypes has been proposed in order to resolve clinical uncertainties still present in the clinical management of patients. Additionally, the prognostic stratification systems have been profoundly modified over the last decade, with a view to refine patients’ staging and being able to choose a clinical approach tailored on single patient’s needs. Here, we will briefly discuss the recent changes in the clinical management of thyroid nodules, and review the current staging systems of thyroid cancer patients by analyzing promising clinicopathological features (i.e., gender, thyroid auto-immunity, multifocality, PTC histological variants, and vascular invasion) as well as new molecular markers (i.e., BRAF/TERT promoter mutations, miRNAs, and components of the plasminogen activating system) potentially capable of ameliorating the prognosis of PTC patients.

## 1. Introduction

Thyroid cancer (TC) derived from the follicular thyroid cell represents the principal endocrine malignancy. It occurs more commonly in women than in men, and it is the fifth most common cancer in the female population of the United States [1,2]. Over the last two decades, the age-standardized incidence rate showed an upward trend, mainly because of the improved ability to diagnose malignant transformation in small non-palpable thyroid nodules [3,4,5]. Most of the epithelial TC have a histologically differentiated phenotype (DTC) and are denoted as differentiated papillary (PTC) and follicular (FTC) TC, which are thought, following dedifferentiation, to give rise to the more aggressive poorly DTC (PDTC), and the incurable anaplastic TC (ATC) [6,7,8,9,10]. Although derived from the same cell type, different TC show definite histological features, biological activities, and degree of differentiation as a consequence of peculiar genetic modifications [6,11]. Established risk factors for TC include radiation exposure, family history of TC, lymphocytic thyroiditis, reduced iodine intake, and female gender [12,13]. Earlier studies, performed on PTC, the most common thyroid malignancy, identified about 70% of the driver mutations implicated in thyrocyte malignant transformation [14,15,16,17,18,19]. The most frequent of these included activating mutations of genes encoding for proteins involved in the mitogen-activated protein kinase (MAPK) signaling pathway (i.e., BRAF and RAS genes), or fusions involving RET and NTRK1 genes [14,15,16,17,18,19]. The progression from DTC to the more aggressive PDTC and ATC is triggered by the occurrence of additional mutations, such as those of the p53 and the telomerase reverse transcriptase (TERT) genes [20,21,22,23]. As for other types of solid cancers, the genetic instability is thought to represent the driving force by which transformed thyrocytes accumulate additional gene mutations during disease progression [24,25]. In fact, a sequential increase in chromosomal abnormalities was observed from well-differentiated PTC to PDTC and ATC, in terms of both number and frequency of detectable abnormalities [24]. Over the last years, with the advent of the next generation sequencing that allowed straightforward investigation of the whole genome, a great advance in the comprehension of the molecular pathogenesis underlying PTC progression has been made [11]. In particular, The Cancer Genome Atlas (TGCA) Research Network identified new oncogenic drivers and new driver events in known cancer genes, and considerably extended the somatic genetic landscape of PTC [11]. This further information was used to propose a reclassification of PTC into molecular subtypes with the aim to improve PTC staging and clinical management [11].

## 2. Thyroid Nodules

Thyroid nodules are very common in the adult population, with a prevalence of 19% to 67% [26,27,28]. Most of them are clinically silent, and only 5% harbor a malignant lesion [26,27,28]. Hence, ruling out malignancy by means of ultrasonography (US) and fine-needle aspiration cytology (FNAC) is the main task in their clinical management [26,27,28]. After the initial assessment of the presence of TC risk factors, US represents the first imaging technique in thyroid nodule evaluation, in that it detects a number of nodule characteristics known to be associated with malignant lesions [29]. These include internal structure of the nodule (solid, mixed, or cystic), echogenicity (hyper-, iso-, hypoechoic or markedly hypoechoic), nodule margins (regular, microlobulated, irregular), presence of calcifications (micro- or macrocalcifications), and shape (taller-than-wide or wider-than-tall) [29]. The US parameters are included in the Thyroid Imaging Reporting and Data System (TI-RADS) score, which shows, compared to the single ultrasound features, greater accuracy in identifying suspicious nodules to be further evaluated by fine needle aspiration cytology (FNAC). To date, FNAC remains the gold standard technique in the evaluation of both palpable and non-palpable thyroid nodules [25,26,27,28]. However, in recent years US evaluation of thyroid nodules has been greatly improved with the introduction of new US software, such as the contrast-enhanced ultrasound (CEUS) and US-elastography (USE) [29]. In particular, USE has emerged as a valuable tool to discriminate malignant from benign nodules, with diagnostic accuracy greater than TI-RADS, and it should become an additional tool for evaluating thyroid nodule differentiation in combination with conventional US and FNAC, as indicated by the World Federation for Ultrasound in Medicine and Biology (WFUMB) and the European Federation of Societies for Ultrasound in Medicine and Biology (EFSUMB) [29,30,31,32]. Moreover, new elastic imaging technologies, such as the shear-wave elastography and the strain ratio elastography, seem to be more efficient in characterizing thyroid nodules reported as indeterminate in FNAC, and their use is expected to increase in the coming years [33]. As mentioned, at present, FNAC represents the gold standard in the diagnosis of thyroid nodules because of its accuracy, reproducibility, and cost effectiveness [25]. Specifically, FNAC-based diagnosis of thyroid tumors shows a sensitivity ranging from 65% to 98%, specificity of 72–100%, and accuracy of 84–95% [26,34,35,36,37]. The Bethesda System for Reporting Thyroid Cytopathology (BSRTC) classifies the FNAC outcome in 6 diagnostic categories including: (1) non-diagnostic; (2) benign; (3) atypia/follicular lesion of undetermined significance (AUS/FLUS); (4) follicular neoplasm or suspicious for follicular neoplasm (FN/SFN); (5) suspicious for malignancy; (6) malignant [38]. Among these, AUS/FLUS and FN/SFN represent a grey zone in which the cytology cannot accurately discriminate malignant from benign lesions, inasmuch they exhibit a malignancy risk of 5–15% and of 15–35%, respectively, based on histological outcome. Thus, a considerable number of patients might undergo unnecessary thyroid surgical procedures [26,39,40]. In this context, the great advance in the comprehension of the molecular pathogenesis of thyroid cancer progression has led to the generation of new molecular approaches capable of ameliorating the diagnostic accuracy of FNAC alone, and to support therapeutic decisions [41,42,43,44,45,46]. In particular, two distinct molecular tests to evaluate FNA samples have entered clinical practice for the management of thyroid nodules [42,44]. The first one is the ThyroSeq v3 genomic classifier; it analyzes 112 genes seeking for more than 12,000 mutation hotspots and 120 fusion types most frequently present in DTC. It is designed to differentiate benign from malignant lesions and possesses a very high predictive value for malignancy [42,43]. The second molecular approach, named gene-expression classifier (GEC) (Veracyte Afirma GSC), was designed to identify benign, rather than malignant, nodules, through the analysis of the expression level of 167 genes in the RNA extracted from FNA biopsies [43,44]. From both tests, a negative result effectively refines the risk of malignancy of AUS/FLUS and FN/SFN diagnostic categories to about 3–4%, comparable to that observed for a benign BSRTC diagnosis [45,46]. Thus, the evaluation and clinical management of thyroid nodules with indeterminate cytology (AUS/FLUS and FN/SFN) should comprise, besides cytology, clinical (i.e., personal or family history of thyroid cancer, lesion size, US features, and elastography) and possibly molecular information. This could lead to a reduction in the number of diagnostic thyroidectomy positively affecting patient’s quality of life [28,43,44,45,46]. It is finally worth considering that the accurate diagnosis of suspicious cervical lymph node (CLN) metastasis is also of great importance in guiding the primary surgical approach, as well as the prognostic stratification and follow-up of TC patients [28,47,48]. As for thyroid nodules, CLN are also evaluated by means of FNAC. However, this task could be challenging as CLN may be a metastatic site of different extrathyroidal malignancies or be affected by diverse non-tumor illnesses [49,50]. In addition, poor cellularity or non-representative sampling, especially in presence of cystic lymph nodes, preclude diagnosis in approximately 20% of cases [51,52]. Thyroglobulin protein and/or mRNA detection in the washout of fine-needle aspirates from CLN was shown to ameliorate the diagnostic accuracy of FNAC alone [28,53,54,55,56,57,58,59]. Furthermore, the same technique may be employed to detect calcitonin, protein, and/or mRNA in case of suspicious metastatic CLN from medullary thyroid carcinomas [28,59].

## 3. Thyroid Cancer Therapy

Thyroid surgery represents the first line therapeutic approach for DTC patients [60,61,62]. Although conventional open thyroidectomy is still the main intervention, over the last decades alternative surgical procedures have been established in order to achieve more pleasing cosmetic results, especially in young women worried about the neck scar [60]. These include the minimally invasive video-assisted thyroidectomy (MIVAT), introduced in the late 1990s, the robot-assisted transaxillary thyroidectomy (RATT), first reported in 2001, and the most recent transoral endoscopic thyroidectomy with vestibular approach (TOETVA) [63,64,65,66]. The surgical treatments of DTC have changed following the introduction of new guidelines by the American Thyroid Association (ATA) in 2015, and by the National Comprehensive Cancer Network (NCCN) in 2018 [26,28,67]. While in the past total thyroidectomy (TT) followed by adjuvant therapy with ^131^I was the treatment of choice for most DTC patients, according to the new guidelines thyroid lobectomy may be offered to low-risk DTC patients [28,68]. More specifically, lobectomy is considered sufficient for patients having small (<1 cm), unifocal, intrathyroidal carcinomas in the absence of prior head and neck radiation, familial history of thyroid carcinoma, or clinically detectable CLN metastases. In addition, some patients with thyroid cancer size ranging from 1 cm and 4 cm without extrathyroidal extension and without any detectable CLN metastases may be selected for lobectomy. In cases where it represents an adequate treatment, unilateral ablation offers the benefits of avoiding hypoparathyroidism, and a lower risk of iatrogenic lesion of the recurrent nerve and permanent chordal paralysis with dysphonia [69]. Following TT, radioactive iodine (RAI) adjuvant therapy is recommended for ATA high-risk patients and should be considered for ATA low to intermediate-risk patients by evaluating individual cases. After RAI remnant ablation or treatment, a whole-body scan should be performed to inform disease staging and document the RAI avidity of any residual disease [26]. Thyroid hormone replacement therapy should be provided in order to maintain TSH level below 0.1 mU/L in high-risk PTC patients, and between 0.1–0.5 mU/L in intermediate-risk PTC patients. For low-risk patients, TSH level can be in the range 0.5–2 mU/L [26,70]. Afterwards, patient follow-up consists of periodic US evaluation of the thyroid bed and CLN compartments, and determination of basal and recombinant human TSH-stimulated thyroglobulin serum level [26,28,70].

The increased knowledge of the molecular mechanisms involved in thyroid cancer progression allowed the development of new therapeutic agents targeting specific pathways involved in disease progression, including RET, BRAF, RAS, epidermal growth factor receptor (EGFR), and vascular endothelial growth factor (VEGF) receptor (VEGFR) [71,72,73]. To date, a number of FDA approved tyrosine kinase inhibitors such as Lenvatinib, Vandetanib, Sorafenib, and Cabozantinib entered clinical practice for the treatment of more aggressive and RAI resistant TC [73,74,75,76,77].

## 4. Thyroid Cancer Patient’s Staging

Although the prognosis of the majority of DTC patients is satisfactory, with 10-years-survival rate of approximately 90%, nearly one third of them face the morbidity of disease recurrences and TC-related deaths [26,28]. The worst outcomes are usually observed in patients with PDTC and ATC, in which the reduced expression of natrium/iodide symporter (NIS) gene renders the RAI treatment less effective or useless. In this context, new hopes arose from a recent study demonstrating that knockdown of STIM1 (stromal interaction molecule 1) in TC cells restored NIS expression and significantly improved iodine uptake, sensitized cells to chemotherapeutic drugs, and significantly reduced xenograft tumor growth [78]. Thus, a precise staging of DTC patients is of crucial importance to ensure the more suitable therapeutic strategy, follow-up, and patients’ quality of life [26]. Different staging systems able to forecast the risk of disease-related death or disease relapse/persistence are accessible [79]. Among these, the most widely employed is the Tumor-Node-Metastasis (TNM) classification elaborated by the American Joint Committee on Cancer (AJCC) [80]. Conceived to predict disease-specific patient’s survival, the TNM staging system remains, however, poorly informative in the prediction of long-term disease outcome [26,81,82,83]. This is because it only incorporates information collected in the period before and immediately after the intervention, and thus provides a rough prediction of cancer mortality over time, grouping in the same risk category patients having very different disease-free and disease-specific survival [28]. In 2009, the American (ATA) and European Thyroid Association (ETA) validated a risk-stratification system for DTC recurrences, in which TNM parameters were implemented by further clinicopathological features such as tumor histology, vascular invasion, radioactive iodine uptake, post-operative thyroglobulin serum level, and others, aimed to distribute patients in three risk-categories: low, intermediate and high [79,84,85]. However, even this revised stratification system has a very low positive predictive value, and individuals included in the same risk group can show very different disease-free intervals (DFI) [86]. In 2015 the ATA substituted the three risk-category model, with a continuum risk model varying from very low to high risk of recurrence [26,86]. In the latter, beside TNM and clinicopathological parameters, mutations of BRAF and TERT genes were included [26,80]. This new stratification system, originally validated in several retrospective single-center studies, has proven to be a reliable predictor of short-term outcomes (12 months follow up after the initial treatment) in a real-world clinical setting by a recent multicenter study enrolling more than 2000 patients with DTC [87]. Hereafter, we will first review the major clinicopathological parameters affecting the current TNM staging system to end with the molecular risk stratification of DTC patients.

### 4.1. Clinicopathological Features Affecting the Prognosis of DTC Patients

As stated above, the TNM staging system elaborated by the AJCC on the basis of clinicopathological features is the most commonly used approach to forecast thyroid cancer survival, but it is much less reliable in discerning patients with higher risk of rising relapses over time [26,80]. The TNM staging has been extensively revised in 2016 (8th edition) in the attempt to achieve a more personalized approach to cancer staging [26,79,87]. This new TNM edition downstages a significant number of patients by increasing the age cut-off from 45 to 55 year and by eliminating the regional lymph node metastases and microscopic extrathyroidal extension from the T3 category [83,86,87,88,89,90]. Although the ability of the new TNM staging to better predict the disease-specific survival (DSS) in DTC patients has been documented, some concerns remain for patients in the 45–54 years range, classified in stages III or IV by the previous TNM classification, but currently considered in stages I or II by the new one, for whom the severity of the disease could be underestimated [88,89,90]. It is worth mentioning that in the latest TNM staging edition it has been suggested to take note, on patient’s record, of molecular data and further clinicopathological parameters in order to evaluate them as potential additional staging factors to include in the next edition [80,88,89,90]. In this context, we next discuss the available clinical evidence supporting the prognostic value of some clinicopathological features for PTC recurrences, i.e., gender, autoimmune thyroid disease (AITD), histological variants, multifocality, and vascular invasion.

Gender—Thyroid cancer is more frequently observed in women. However, males tend to develop more aggressive tumors and to have poorer survival [91,92,93,94,95,96,97,98,99,100,101,102,103], although there is no general agreement on this matter [92,93,94,95,96,97,98,99,100]. Guo and Whang reported in 2014 the results of a meta-analysis performed on 13 studies involving 7048 patients [95]. They concluded that the male sex actually represents a risk factor influencing PTC recurrences [95]. In this regard, the observation of Choi and colleague is of interest, which analyzed 3147 PTC patients treated at the Seoul National University Hospital between 1962 and 2009 [100]. They reported that the risk of poor survival and recurrence associated with male sex decreased over time, while the risk related to other clinicopathological parameters remained the same or increased. These trends might be associated with recent changes in the characteristics of thyroid cancer, especially considering the increasing number of PTC diagnosed as microcarcinomas [3,100]. Moreover, a recent multivariate analysis demonstrated that, although PTC recurrence and death were more common in male than in female patients, there was no difference in DFI between genders in papillary thyroid microcarcinoma (PTMC) [101]. In this study, male gender was not an independent prognostic factor of recurrence in PTMC [93]. Analogous results were obtained in more recent studies [83,102].

AITD—A number of studies described the correlation of AITD, and in particular of chronic lymphocytic thyroiditis (CLT), with PTC [83,104,105,106,107,108,109,110,111]. In one of these, TSH serum level and thyroid autoantibodies were evaluated in 13,738 PTC patients, of whom 3914 under thyroxine treatment and 9824 untreated [106]. The findings showed that the prevalence of PTC was higher in patients affected by CLT and was associated with increased TSH levels [106]. In several studies, both AITD and elevated TSH levels were found to represent independent risk factors for thyroid malignancy [107]. More recently, Moon and colleagues reported the results of a meta-analysis evaluating the effects of CLT on the clinical outcome of PTC patients [108]. The authors examined 71 articles for a total of 44,034 patients, among which 11,132 had CLT. They observed a negative association between CLT and extrathyroidal extension, lymph node metastases, distant metastases, and disease recurrence [108]. This evidence was further corroborated by a recent study performed on 2070 PTC patients, showing that those who were positive for thyroid peroxidase antibodies (TPOAb) before surgery had a significantly longer DFI compared to patients negative for TPOAb [109]. In addition, the presence of preoperative TPOAb was found to be an independent prognostic factor of persistent/recurrent disease after adjustment for major preoperative risk factors such as age at diagnosis, gender, and tumor size [111]. In addition, Myshunina and colleagues reported that the presence of chronic thyroiditis in DTC patients had a positive impact on the course of the disease [112]. In particular, DTC patients with thyroiditis showed reduced tumor size, invasion of extrathyroidal structures, and lateral CLN metastasis compared to patients without thyroiditis [112]. These observations are in agreement with a suggested protective role of autoimmunity against cancer [113,114,115]. Thus, it appears worthwhile to consider AITD as a possible additional prognostic factor for the prediction of both PTC-specific survival and recurrences.

Histological variants—Several investigations evaluated the prognostic value of the major PTC variants [11,116]. As will be discussed later on, in the framework of The Cancer Genome Atlas (TCGA) project, a comprehensive multiplatform analysis was carried out to determine the genomic landscape of 496 PTC, and a reclassification of PTC into molecular subtypes was proposed in order to improve clinicopathological grading and management of patients [11]. In this study, the lowest thyroid differentiation score (TDS) was assigned to a tall cell-like tumor cluster, which was associated with more advanced stage and higher risk, while the classical PTC (CPTC) had an intermediate TDS, and the follicular variant (FVPTC) maintained a high TDS [11]. These results were corroborated by a subsequent multicenter retrospective study, including 6282 cases of PTC [116]. Differential risk patterns of disease recurrence and patient mortality were determined for the three major PTC variants, with increasing aggressiveness from the FVPTC to the CPTC, up to the tall cell PTC (TCPTC) variant [116]. A significantly worst prognosis associated with the TCPTC variant was recently confirmed on a case study of 1148 PTC patients [83].

Multifocality—Some studies indicated that multifocality is present in about 30–37% of PTC patients [83,117,118,119,120,121,122,123,124,125]. However, whether it may represent an independent risk factor for disease recurrence and overall mortality is still a matter of debate, because different reports produced conflicting results [117,118,119,120,121,122,123,124,126]. In particular, in a large multicenter study performed on 2638 PTC patients, multifocality was not found to be an independent prognostic marker for either PTC recurrence or death [124]. In addition, the lack of prognostic value of multifocality emerged from the analysis of 89,680 PTC patients entered in the Surveillance, Epidemiology, and End Results (SEER) database [124]. Similar results were recently obtained from the largest UK series of PTC collected to date, showing that multifocality was not an independent predictor of outcome on multivariate Cox proportional hazards regression analysis [125]. Nevertheless, multifocality is currently included in the ATA continuous risk scale for PTC relapse risk assessment [26].

Vascular invasion—Vascular invasion (VI) is frequently observed in DTC, and several investigators attempted to evaluate its prognostic role, but the available data are controversial [83,127,128,129,130,131,132]. VI rate is higher in FTC than in PTC, explaining why FTC metastasizes to distant organs more frequently than PTC does. However, the impact of VI on recurrences has not yet been defined. From a recent meta-analysis, which included 26 studies comprising 11,961 DTC patients, a significant association of VI with tumor persistence and worse DSS was evidenced [132]. VI has already been included among the parameters of the ATA continuous risk scale for PTC recurrence and could be considered for integration in the next TNM staging edition [26].

### 4.2. Molecular Risk Stratification of DTC Patients

As mentioned above, the TGCA research network, taking advantage of multiplatform ‘omics’ methodologies such as next-generation DNA and RNA sequencing, copy-number analysis, transcriptomic, methylomic and proteomic assays, performed a comprehensive molecular characterization of 496 PTC and normal thyroid tissues [11]. Following the identification of 71-genes signature panel, PTC were divided into two categories: BRAF^V600E^-like and RAS-like tumors. The former mainly includes the classical and tall cells PTC variants and shows a lower thyroid differentiation score (TDS) and a higher risk of recurrences. This group, according to the different data set analyzed (i.e., proportion of driver mutations, gene fusions, histology, age, TDS, etc.), could be further divided into different non-overlapping subtypes [11]. One of these, namely the tall cell PTC variant (TCV), showed the highest frequency of TCV and BRAF^V600E^ mutations, the lowest TDS, and was associated with more advanced stages and higher risk [11]. On the other hand, the RAS-like PTC mainly include the follicular variant of PTC (FVPTC), and associate with younger patient’s age, high TDS, and relatively low risk of recurrences [11]. Later on, these observations were confirmed by Yoo and colleagues who analyzed the transcriptional and mutational landscape of 180 TC, including 30 minimally invasive FTC (miFTC) and 25 follicular adenomas (FA), along with 77 classical PTC and 48 FVPTC [133]. The analysis of gene expression profiles confirmed the BRAF- and RAS-like molecular subtypes previously identified by TGCA and distinguished a third molecular subtype defined as Non-BRAF-Non-RAS (NBNR), comprising FA and miFTC [133]. At present, however, classification of DTC into 3 molecular groups does not seem to improve the prognostic stratification of patients [134]. Recently, using information on the PTC patients’ cohort available in the TCGA data portal, we sought to verify whether DFI prediction, provided by clinicopathological parameters such as lymph node metastasis or age could be improved by molecular variables such as number of total non-silent mutations, number of CpGT mutations, BRAF-RAF score, ERK score, miRNA and RPPA clusters, ploidy, and TDS [11,83,135,136]. However, none of these was found to be a significant predictor of DFI in the Cox regression model, with lymph node involvement being the best predictor of DFI, and with the N1b category showing the highest odds ratio [83]. On the whole, a number of molecular parameters have been considered in the prognostic stratification of DTC patients over the last couple of decades. The clinical evidence supporting a prognostic value for some of them will be reviewed below, starting from the BRAF^V660E^ and TERT promoter mutations currently included in the ATA prognostic stratification of DTC patients [26].

#### 4.2.1. BRAF^V600E^ Mutation

The BRAF^V600E^ mutation is the most prevalent genetic alteration (up to 80%) found in PTC tissues [11]. However, whether it could represent a reliable prognostic marker is highly debated [134]. Several studies reported an association between BRAF^V600E^ mutation and PTC recurrences, presence of lymph node metastasis, advanced tumor stage and worse prognosis [137,138,139,140,141,142,143]. Xing and colleagues, in a large multicenter study including more than 2000 patients, demonstrated an independent prognostic value of BRAF^V600E^ mutation for PTC recurrences even in patients with low TNM stage and micro-PTC [144]. These observations however, were not confirmed by other studies [145,146]. In a recent meta-analysis, BRAF^V600E^ mutation showed its prognostic value only in short/medium-term follow-up [147]. Moreover, in PTC patients the frequency of BRAF^V600E^ mutation (up to 80%) is high, while the prevalence of a negative outcome (10–15%) is low [132,138]. As a consequence, based only on the analysis of BRAF mutation a considerable number of patients would face the risk of over- or undertreatment. Thus, BRAF^V600E^ mutation should be considered one of the factors influencing the prognosis of PTC patients, but it should be evaluated together with other prognostic factors [145,146,148,149,150].

#### 4.2.2. TERT Promoter Mutations

Unrestricted cell proliferation, along with telomerase activation, represent well-known hallmarks of cancer, including TC [151,152]. Mutations of the promoter region of the TERT gene (TERTp) have been shown to occur in several cancer types [153]. They arise in two hot spots of the TERTp and are thought to generate E-twenty-six/ternary complex factors (Ets/TCF) consensus binding sites (GGAA) conferring enhanced TERT promoter transcriptional activity [154,155]. These mutations have a low prevalence in PTC but strongly associate with aggressive forms, being highly frequent (up to 50%) in PDTC and ATC [156,157]. Actually, TERTp mutations have been related to poor clinical outcomes of TC characterized by increased recurrence rate and disease related mortality [158,159,160,161,162,163,164]. Such evidence demonstrates a primary oncogenic role of TERTp mutation in the progression towards the most aggressive TC.

#### 4.2.3. The Connection BRAF^V600E^-TERTp Mutations

Since their initial identification in TC, it was evident that TERTp mutations, besides being more common in the most aggressive TC (i.e., TCPTC, PDTC, ATC), are associated with BRAF^V600E^ PTC [20]. In particular, the TERTp mutation C228T showed a significant higher prevalence in BRAF^V600E^ PTC (18.3%), compared to BRAF wild-type PTC (7.2%). Knowing the oncogenic role of BRAF^V600E^ and TERTp mutations, it was soon demonstrated that the concurrent presence of the two mutations had a strong prognostic value for aggressive TC [20,155,159,160,162,163,164]. These data were taken into consideration by the ATA, which incorporated the BRAF and/or TERTp status in the continuous risk scale for PTC relapse risk assessment [26].

#### 4.2.4. MicroRNAs

MicroRNAs (miRNAs) are versatile regulators of gene expression in higher eukaryotes. They consist of short (~17–22 nt) single-stranded ribonucleic acids able to bind their target mRNAs, usually in their 3′ untranslated regions (UTRs), inhibiting translation or inducing mRNA degradation or deadenylation (Figure 1) [165]. As reported in Figure 1, miRNA biogenesis is a multistep process. The primary transcripts (pri-miRNA) generated by the RNA polymerase II or III are first cleaved by the concerted action of the nuclear endoribonuclease Drosha and the ds-RNA binding protein DGCR8 (DiGeorge critical region 8). The released pre-miRNA is then exported to the cytoplasm by the Exportin-5 (Exp5) where the endoribonuclease Dicer generates a ~17–22 nt RNA duplex, which is bound by the Argonaute (Arg) protein. The complementary target mRNA strand is selected to form the mature miRNA effector as part of a miRISC (miRNA-induced silencing complex), while the remaining strand is degraded (Figure 1) [165]. In physiological conditions miRNAs operate a temporally and tissue-specific controlled post-transcriptional regulation of gene expression, and not surprisingly their dysregulation was shown to be involved in cell malignant transformation owing to the downregulation of tumor suppressor genes and/or upregulation of oncogenes [166,167]. A number of miRNA genome-wide studies, performed in different human malignancies, evidenced the presence of selective groups of distinct miRNAs (miRNA fingerprints) dysregulated in specific tumor types [165]. From these studies, the clinical utility of miRNA expression determination in cancer tissues also emerged, as they were found to be significantly associated with diagnosis, prognosis, and response to clinical therapies [11,166]. Several studies attempted to characterized miRNA profile in TC, and the most consistently upregulated miRNAs were found to be the miR-146b, miR-221, and miR-222 [11,167,168]. The levels of miR-146b were reported to be significantly higher in PTC tissues with extrathyroidal invasion, and to have a direct correlation with tumor size and higher TNM stage [169]. Chou and colleagues demonstrated that PTC patients with higher miR-146b expression were characterized by worst overall survival [170]. In addition, they reported that its overexpression increased cell migration and invasiveness, and resistance to chemotherapy-induced apoptosis. In multivariate analysis, miR-146b was shown to represent, along with advanced tumor stage and cervical lymph node metastases, an independent risk factor for poor prognosis in PTC [170].

These observations were corroborated by the findings of the TGCA network, reporting a high miR-146b expression in PTC tissues and its correlation with DNA methylation, BRAFV600E-RAS score, and TDS [11]. The miR-221 and miR-222, encoded on chromosome X (Xp11.3), share the same seed sequence and thus are thought to influence the same target mRNAs, including the cell cycle regulator p27 [169,171,172]. They are often deregulated in different malignancies, and in PTC they are both upregulated and associated with poorly differentiated tumor tissues [11]. Mardente and colleagues demonstrated, in primary cultures and cell lines derived from PTC, that miR-221 and miR-222 overexpression entailed increased cell growth and motility [172]. The prognostic value of miR-221 and miR-222 was recently evaluated in a large meta-analysis, including fifty studies for a total of 6086 patients with different non-thyroid cancer types [169]. From this work, it emerged that high miR-222, but not miR-221 expression, represents a biomarker of poor prognosis in terms of both overall survival and secondary outcomes, e.g., DFI [169]. These findings should warrant further studies aiming to assess the prognostic relevance of miR-221 and miR-222 in TC patients. In a very recent study, Akyay and colleagues analyzed the global transcriptome and miRNA profile of three groups of PTC tissues: PTC localized to the thyroid; PTC with extrathyroidal progression; PTC with distant metastasis [173]. By comparative analysis of differentially expressed miRNAs, the authors identified the miR-193-3p, miR-182-5p, and miR-3607-3p as associated with PTC metastasis, suggesting that they could serve as new biomarkers for the identification of PTC patients at risk of disease progression or metastatization [173]. Besides those described, a number of other miRNAs have been shown to be deregulated and potentially capable of affecting TC progression, including miR-181p, miR-182, miR-183, miR-204, miR-206, miR-128-3p, miR-375, and others [11]. For these, more extensive and in-depth investigations aimed to clarify their potential prognostic value are needed.

#### 4.2.5. Components of the Urokinase Plasminogen Activating System

The urokinase plasminogen activating system (uPAS) includes the urokinase plasminogen activator (uPA), the plasminogen activator inhibitors 1 (PAI-1) and 2 (PAI-2), and the uPA cell membrane receptor (uPAR) (Figure 2) [174]. It is involved in many physiological and pathological processes, including wound healing, tissue regeneration, angiogenesis and, along with the matrix metalloproteases (MMPs), extracellular matrix (ECM), and basement membrane (BM) remodeling [174]. A number of observations documented the ability of the uPAS to affect several malignant cell features, including proliferation, migration, adhesion, intravasation and extravasation and tumor neoangiogenesis, and to play a prominent role in cancer invasion and metastatization (Figure 1) [174,175]. In addition, high tumor tissue levels of one or more uPAS components associate with poor prognosis in several human malignancies [174]. This is particularly evident in breast cancer, where uPA and PAI-1 represent potent prognostic factors, with a predictive value stronger than those of patient age, tumor size, estrogen, and progesterone receptors, HER-2/neu, or p53 expression [175,176]. Actually, evaluation of uPA and PAI-1 protein levels in extracts of breast cancer tissue is recommended to identify patients prone to a worse outcome [177].

Experimental evidence documented an increased expression of uPAS components during TC progression [174,178]. An early study reported the association of high uPAR expression in PDTC, indicating this protein as a putative biomarker [179]. A subsequent study showed that the levels of uPA and PAI-1 proteins displayed the lowest values in adenomas and the highest in anaplastic carcinomas [180]. Among DTC, uPA and PAI-1 were found higher in tumors with extrathyroidal invasion or distant metastasis. More interestingly, the survival analysis revealed a significant impact of both uPA and PAI-1 on the progression-free survival rate [180]. An increased uPA, uPAR, and PAI-1 expression was also reported in TC-derived cell lines and PTC tissues compared to normal human thyrocyte cultures or matched normal thyroid tissues, respectively [181]. In addition, a correlation was found between tumor size and uPA expression, and higher levels of uPA and uPAR were detected in metastatic PTC [181]. These observations were corroborated by another study, showing the association between uPAR and disease-specific survival in a case study encompassing PTC, MTC, FTC, and ATC patients [182]. Later on, our group demonstrated that the increased gene expression of uPA and uPAR in PTC tissues was associated with tumor invasiveness, advanced stages, and shorter DFI, and that this association was even stronger in TNM stage I patients, currently considered at low risk of recurrences [145,183]. Finally, a significantly higher uPA and uPAR expression in BRAF^V600E^-positive PTC was also reported, compared to those bearing the wild type BRAF [145]. On the whole, these findings clearly indicate a correlation between the increased expression of one or more uPAS components and a worst prognosis in TC patients, which should encourage larger case studies to validate the prognostic value of uPAS in TC.

#### 4.2.6. Other Possible Biomarkers

Several experimental and clinical data suggest that other molecules may have a role in the prognostic stratification of TC patients. These include immune checkpoint factors and immune-related signature (IRS), estrogen receptors, Vitamin D, tumor angiogenic microenvironment, circular RNAs, long noncoding RNAs, and the angiotensin converting enzymes ACE and ACE2 [184,185,186,187,188,189,190,191,192,193,194]. However, all of them are awaiting validation in large case studies.

## 5. Conclusions and Perspectives

The characterization of the molecular pathogenesis underlying thyroid cancer progression is expected to shed light on some critical issues still present in the clinical management of thyroid cancer patients. To date, the molecular diagnosis has been shown to ameliorate the diagnostic performance of fine-needle aspiration cytology, reducing the number of unnecessary thyroidectomies and improving quality of life in a significant number of patients. In addition, the inclusion of TERTp mutations in tumor staging, alone or in combination with BRAF^V600E^, is emerging as a useful tool in refining the prognostic classification of DTC patients. It is likely that in the near future, the TNM classification systems will be further implemented with additional clinicopathological and molecular features, actually under evaluation, capable of ameliorating the prognosis of these patients.

## Figures and Tables

**Figure 1 cancers-13-05567-f001:**
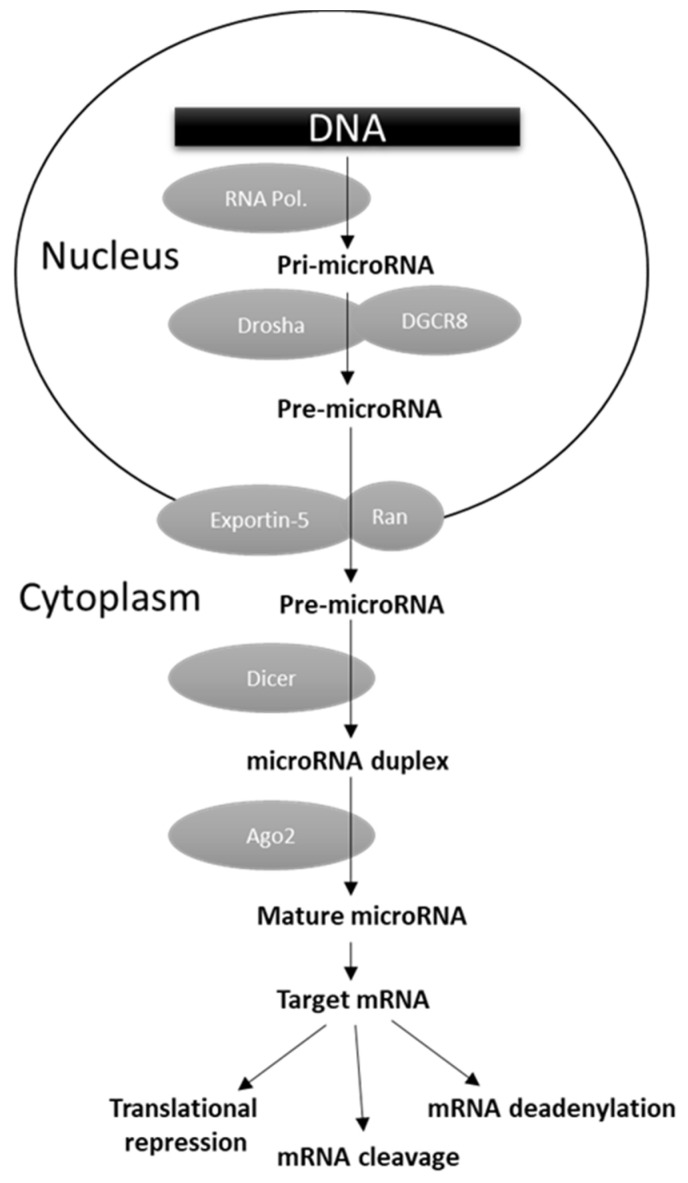
miRNA biogenesis and function.

**Figure 2 cancers-13-05567-f002:**
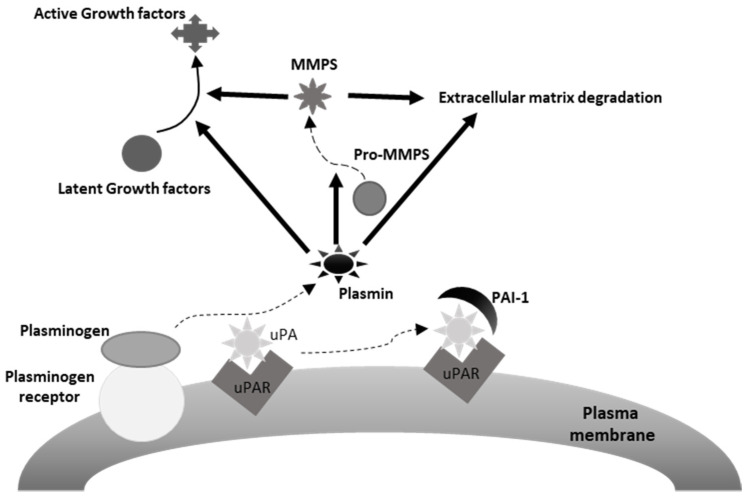
Depiction of the urokinase plasminogen activating system. uPA, urokinase plasminogen activator, uPAR, uPA receptor, PAI-1, plasminogen activator inhibitor-1. As it may be appreciated in the figure, plasmin induces extracellular matrix degradation directly and indirectly, through the activation of extracellular matrix bound MMPs (matrix metalloproteinases). Similarly, plasmin may activate latent mitogenic and motogenic growth factors associated with the extracellular matrix, thus promoting tumor cell proliferation and invasion.

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
