# Peer review of "Papillary Thyroid Cancer Prognosis: An Evolving Field"

_cancers, 2021, doi:10.3390/cancers13215567_

Round 1

Reviewer 1 Report

In this paper the authors present a review of prognostic systems for papillary thyroid cancer. The authors present an overview of thyroid cancer management and the paper is informative and contains some interesting references. I have concerns with one section:

The section on thyroid cancer therapy misrepresents the ATA guidelines which state that either total thyroidectomy or lobectomy can be considered for patients with a PTC over 1cm and not that total thyroidectomy should only be offered to high risk patients. The claim that patients ‘Patients lacking these high-risk factors should undergo thyroid lobectomy, to eventually a secondary complete thyroidectomy only in case that high risk factors, such as aggressive phenotype emerge from the histological evaluation [28,67,68]’ misrepresents the ATA guidelines and if this is the authors individual practice this should be stated and not presented as fact and it should be stated that high quality evidence is lacking to support this approach which requires further research. The reference number 68 cited to support this comment presents a retrospective case series and does not appear to be relevant to the clinical approach the authors are proposing.

Similarly the ATA guidelines state that ALL patients with intermediate risk should be considered for radioactive iodine not selected cases – the ATA guidelines state on page 55 that ‘RAI adjuvant therapy should be considered after total thyroidectomy in ATA intermediate-risk level DTC patients’. In addition the sentence on thyroid cancer follow up is misleading with regards to whole body scans which the authors should review recommendation 66 on page 69 of the guidelines.

The messages from this section are concerning and in my opinion present a simplified and incorrect reading of the literature and this section needs to be completely revised if this paper is to be published.

The remainder of the paper in my opinion presents a relevant review of thyroid cancer biology and discussion of translational research.

Author Response

We are grateful to the Reviewer for his/her work and comments on the manuscript and for evidencing the discrepancies between what reported in the “Thyroid cancer therapy” paragraph of the original manuscript and the ATA guidelines. In agreement with the Reviewer’s advices the paragraph has been modified in the revised manuscript (see end of page 3 and page 4) and the original reference 68 eliminated. Also english presentation has been revised. Note that all changes are reported in bold in the revised manuscript.

Reviewer 2 Report

In this manuscript titled “Papillary Thyroid Cancer Prognosis: An Evolving Field”, the authors have reviewed actual staging system of thyroid cancer patients and discussed most relevant clinicopathological parameters and new molecular markers potentially capable to refine the prognosis of differentiated thyroid cancer (DTC).

The manuscript is well written and provides an extensive and up to date review of literature.

I suggest following minor changes that would further improve the manuscript.

1. In Section, Thyroid Cancer Therapy: the authors may briefly (in couple of sentences) add the targeted drugs therapy by several multi kinase inhibitors including Lenvatinib, Sorafenib, Cabozantinib, doxorubicin.

2. In Section, Thyroid cancer patient’s staging: The authors stated that In poorly differentiated thyroid cancer (PDTC) and anaplastic thyroid cancer (ATC), the expression of NIS is decreased which leads to minimal or no iodine uptake by the thyroid cancer cells.

Authors may consider citing here a recent study that the knockdown of stromal interaction molecule 1 (STIM1) in aggressive thyroid cancer cells restores the NIS expression and significantly improves iodine uptake by the cancer cells:

Asghar MY, Lassila T, Paatero I, Nguyen VD, Kronqvist P, Zhang J, Slita A, Löf C, Zhou Y, Rosenholm J, Törnquist K. Stromal interaction molecule 1 (STIM1) knock down attenuates invasion and proliferation and enhances the expression of thyroid-specific proteins in human follicular thyroid cancer cells. Cell Mol Life Sci. 2021 Aug;78(15):5827-5846. doi: 10.1007/s00018-021-03880-0. Epub 2021 Jun 21. PMID: 34155535; PMCID: PMC8316191.

  1. Authors may consider adding a comprehensive table for section 4.

Author Response

We are grateful to the reviewer for his/her work and comments on the manuscript. We have revised the manuscript according to the criticisms stated and corrected the english presentation. Note that all changes are reported in bold in the revised manuscript. Reply to specific comments are as follow:

  1. The potential diagnostic and/or prognostic role of angiotensin converting enzymes, ACE and ACE2, has been included in the section “Other possible biomarkers” reported at page 11 of the revised manuscript, and the suggested reference included (Reference 189 of the revised manuscript).
  2. The diagnostic value of shear wave elastography and strain ratio elastography has been acknowledge in the revised manuscript (see end of page 2) and the suggested reference included (reference 33 of the revised manuscript).
  3. As suggested by the Reviewer, thyroid surgery complications have been reported in the revised manuscript (see end of page 3 and beginning of page 4), and the suggested reference included (reference 69 of the revised manuscript).

Reviewer 3 Report

Only minor corrections:

- page 1, line 6, Angiotensin converting enzymes, ACE and ACE2, not only play a fundamental role in blood pressure regulation, but are involved in pathophysiological processes, including thyroid dysfunction or the progression of various neoplasms such as cancers of the skin, lungs. , pancreas and leukemia. However, their role in thyroid carcinogenesis remains unknown. It was shown that ACE was significantly decreased in individuals over the age of 50. Both ACEs were significantly decreased in M1 patients, ACE2 also in higher tumor masses. ACE and ACE2 are regulated within benign and malignant thyroid tissues. Since the transcription ratio between both enzymes is proportional to the differentiation status of thyroid cancer, ACE and ACE2 can serve as novel markers for thyroid cancer.. please cite doi:10.4149/neo_2019_190506N405

  • page 2, line 3, The diagnostic reliability of elastography in thyroid nodules reported as indeterminate in previous fine-needle aspiration cytology (FNAC) is a debated topic, although elastography has had discrete diagnostic results in indeterminate thyroid nodules. Shear wave elastography and strain ratio elastography could be more efficient in diagnosis and are expected to evolve in the coming years, while the combination of elastography with ultrasound would currently contribute more to sensitivity and specificity. please cite doi:10.1007/s00330-020-07023-0
  • page 4, after and anesthesia risks [68]. thyroid surgery to date still remains burdened by heavy complications such as iatrogenic lesion of the recurrent nerve and permanent chordal paralysis with dysphonia. please cite doi:10.2217/fon-2019-0053

Author Response

We are grateful to the reviewer for his/her work and comments on the manuscript. The manuscript it has been revised according to Reviewer’s advice and english presentation revised. All changes made are reported in bold in the revised manuscript. Reply to specific comments are as follow:

  1. Following Reviewer’s advice, a small paragraph indicating the new targeted therapies has been added at the end of the “Thyroid Cancer Therapy” section.
  2. The information has been added including the suggested reference (Ref. 78 of the revised manuscript).
  3. Unfortunately, we do not agree with the Reviewer on this point as the insertion of a table in section 4 will not add much value to the manuscript. Thus, we apologize with the Reviewer, but we decided not to include it.